# Human-to-Cat SARS-CoV-2 Transmission: Case Report and Full-Genome Sequencing from an Infected Pet and Its Owner in Northern Italy

**DOI:** 10.3390/pathogens10020252

**Published:** 2021-02-23

**Authors:** Gabriele Pagani, Alessia Lai, Annalisa Bergna, Alberto Rizzo, Angelica Stranieri, Alessia Giordano, Saverio Paltrinieri, Davide Lelli, Nicola Decaro, Stefano Rusconi, Maria Rita Gismondo, Spinello Antinori, Stefania Lauzi, Massimo Galli, Gianguglielmo Zehender

**Affiliations:** 1Infectious Diseases Unit, 3rd Division, Luigi Sacco Hospital, ASST FBF-Sacco, 20157 Milan, Italy; stefano.rusconi@unimi.it (S.R.); spinello.antinori@unimi.it (S.A.); massimo.galli@unimi.it (M.G.); 2Luigi Sacco Department of Biomedical and Clinical Sciences, Università Statale di Milano, 20157 Milan, Italy; alessia.lai@unimi.it (A.L.); annalisa.bergna@unimi.it (A.B.); gianguglielmo.zehender@unimi.it (G.Z.); 3Microbiology Unit, Luigi Sacco Hospital, ASST FBF-Sacco, 20157 Milan, Italy; alberto.rizzo@unimi.it (A.R.); mariarita.gismondo@unimi.it (M.R.G.); 4Department of Veterinary Medicine, University of Milan, 26900 Lodi, Italy; angelica.stranieri@unimi.it (A.S.); alessia.giordano@unimi.it (A.G.); saverio.paltrinieri@unimi.it (S.P.); stefania.lauzi@unimi.it (S.L.); 5Istituto Zooprofilattico Sperimentale della Lombardia e dell’Emilia Romagna (IZSLER), 25124 Brescia, Italy; davide.lelli@izsler.it; 6Department of Veterinary Medicine, University of Bari, 700010 Bari, Italy; nicola.decaro@uniba.it

**Keywords:** SARS-CoV-2, cat, full-genome analysis, one health

## Abstract

There have been previous reports of the human-to-cat transmission of SARS-CoV-2, but there are only a few molecular studies that have compared the whole genome of the virus in cats and their owners. We here describe a case of domestic SARS-CoV-2 transmission from a healthcare worker to his cat for which nasopharyngeal swabs of both the cat and its owner were used for full-genome analysis. The results indicate that quarantine measures should be extended to pets living in SARS-CoV-2-infected households.

## 1. Introduction

Susceptibility to SARS-CoV-2 infection has been demonstrated in a wide range of mammals under laboratory conditions, with cats and ferrets being the most permissive hosts [1]. Cat-to-cat transmission has also been demonstrated in experimentally inoculated animals [2], and it is known that the various outbreaks of SARS-CoV-2 infection on mink farms in The Netherlands and Denmark have been caused by human-to-animal, mink-to-human, and mink-to-cat transmission [3,4,5,6]. Furthermore, although the transmission of SARS-CoV-2 is mainly between humans, there have been reports of probable human-to-cat transmission [7,8], including transmission from infected owners of pet cats [9,10]. In Italy, recent testing of oropharyngeal, nasal and/or rectal swabs of about 900 pets revealed that the prevalence of anti-SARS-CoV-2 antibodies among the cats was 5.8%, while no RT-PCR positive swab was found [11].

Given the possible presence of intermediate hosts and animal reservoirs, understanding the dynamics of SARS-CoV-2 transmission and controlling the current COVID-19 pandemic requires a One Health approach [12] and detailed knowledge of viral circulation in humans and animals. However, only a few of the published reports have included an in-depth analysis of viral genomic sequences both in the infected cats and their owners [9,10].

The aim of this paper is to describe a case of the human-to-cat transmission of SARS-CoV-2 and provide a full-genome analysis of the viruses infecting the cat and its owner.

## 2. Case Report and Virological Analyses

A resident working at an Infectious Diseases Unit of L. Sacco Hospital (Milan, Italy) reported upper respiratory symptoms and severe bilateral coxo-femoral arthralgia that started on 18 March, lasted 3–4 days, and was followed by long-lasting olfactory disorders. No fever, dyspnea, or the other signs or symptoms of COVID-19 were reported and so, in accordance with the Regional Health Protocol at the time, he did not undergo nasopharyngeal swab (NPS) real-time RT-PCR testing for SARS-CoV-2 RNA.

On the 6 April, a rapid immunochromatographic lateral flow test (PRIMA Lab SA, Balerna, Switzerland) was weakly positive for SARS-CoV-2 IgG antibodies, and an NPS collected on the following day was tested for SARS-CoV-2 (Liferiver™ Novel Coronavirus (2019-nCoV) Real-Time Multiplex RT-PCR Kit). The result was “weakly positive”, with cycle thresholds [CT] for genes N, E and RdRP of respectively 36, 35 and 37. Subsequent NPSs collected on 12 and 13 April were both RT-PCR negative (Figure 1)

The presence of frequent sneezing in the resident’s 4-year-old short-haired cat, named Zika, induced him to take an oropharyngeal swab (OPS) on the 13th of April even though the cat did not appear ill and was eating regularly. Real-time RT-PCR (CLONIT Quanty COVID 19 CE-IVD, Clonit Srl., Italy) revealed that the swab was positive, with a CT of 34. A second OPS and a rectal swab (RS) taken a week later were both negative (Figure 1).

The SARS-CoV-2-positive OPS was used to isolate the virus on Vero E6 cells [13]. The supernatant of the infected cell culture was RT-PCR positive with a CT of 28 after a week of incubation.

The whole genomes of the human and feline strains were obtained by means of next-generation sequencing and the Swift Amplicon^®^ SARS-CoV-2 panel (Swift Biosciences, Ann Arbor, MI, USA) using the 2 × 150 cycle paired-end sequencing protocol on the Illumina MiSeq platform. The results were mapped and aligned with the reference genome obtained from GISAID (https://www.gisaid.org/ (accessed on 29 May 2020), accession ID: EPI_ISL_412973) using Geneious software, v. 9.1.5 (http://www.geneious.com (accessed on 29 May 2020)).

The strains showed 99.9% nucleotide identity, and both belonged to the Nextstrain clade 19A (https://clades.nextstrain.org/ (accessed on 29 May 2020)), corresponding to the B lineage of Pangolin (https://pangolin.cog-uk.io/ (accessed on 29 May 2020)). A maximum-likelihood tree was generated using the Italian dataset on IQTree V.2, and phylogenetic analyses confirmed the clade classification and indicated that the two sequences significantly grouped in the same cluster (Figure 2).

Table 1 shows the amino acid mutations in the encoded proteins of the two sequences. Both genomes showed previously reported mutations, including the D614G mutation in the Spike glycoprotein, and mutations that have not been previously described; the human viral sequence had additional mutations not found in the cat strain.

During the follow-up, the owner did not notice any sign of illness in the cat, and a veterinary examination about six months after the first positive OPS confirmed the absence of any clinical abnormalities. During the examination, a blood sample was taken in order to make a complete blood count and biochemical analysis, including serum protein electrophoresis, the results of which were unremarkable. A virus neutralisation test of the same blood sample [14] was negative for SARS-CoV-2 antibodies. The OPS and RS collected during the same veterinary examination were real-time RT-PCR negative for viral genome.

## 3. Discussion

The presence of SARS-CoV-2 in a cat four weeks after the onset of clinical signs in its COVID-19 positive owner suggests human-to-cat transmission, especially as the cat was kept indoors and had had no contact with anyone other than its owner during the previous two weeks, and confirms previous reports [8,9,10,11].

Whole-genome sequencing of the viruses from the infected cat and its owner confirmed their close genetic relationship. Only two genome comparisons of SARS-CoV-2 isolated from cats and their owners have been reported so far [9,10], although it has been suggested that cats may be easily infected by their COVID-19-positive owners [8,9,10,11].

None of the mutations in the cat’s sequence have been previously detected in pets, and the larger higher number of mutations observed in the owner’s sequence may have been due to a longer time of infection in humans and/or less efficient viral replication in cats. The phylogenetic analyses made using viral sequences from Lombardy (Italy) that were collected during the same period as the study sequences highlighted the presence of a single and separate clade that included the strains of the cat and its owner, and showed that these strains belonged to a PANGO lineage (B) that has not been frequently detected in Lombardy [15].

Our findings confirm previous observations of asymptomatic or mildly symptomatic SARS-CoV2 infection in cats [8,9,10,11], although it cannot be excluded that it may lead to severe disease, as in the case of other feline coronaviruses [1].

Unfortunately, we could not determine whether or when seroconversion occurred, nor how long specific antibodies can be found in cats, because no blood sample was taken at the same time as the OPS.

Cat-to-human transmission has never been reported, and the very low frequency of positive cats in COVID-19 households [11] and the fact that the duration of viral shedding is shorter in naturally infected cats than in their owners [16] suggest that viral transmission between domestic cats and humans is unlikely. However, it has been reported that cats can be naturally infected as a result of contact with other animals [2,6]. Further studies are necessary in order to determine whether cats act as virus reservoirs during interepidemic periods and therefore may develop antigenically relevant variants as has been observed in the case of minks [4,5].

In conclusion, the findings of this study should encourage COVID-19-positive cat owners to avoid close contact with their pets in order to prevent virus transmission. The World Organisation for Animal Health (OIE) recommends that domestic cats in COVID-19-positive households should quarantine like their owners, although this is mainly for their well-being and not due to public health concerns [17].

## Figures and Tables

**Figure 1 pathogens-10-00252-f001:**
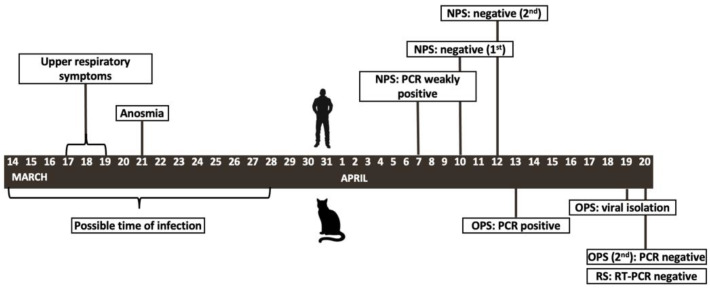
Timing of the diagnostic tests of the cat (**below the timeline**) and its owner (**above the timeline**). The possible time of infections is speculative. NPS: nasopharyngeal swab; OPS: oropharyngeal swab; RS: rectal swab.

**Figure 2 pathogens-10-00252-f002:**
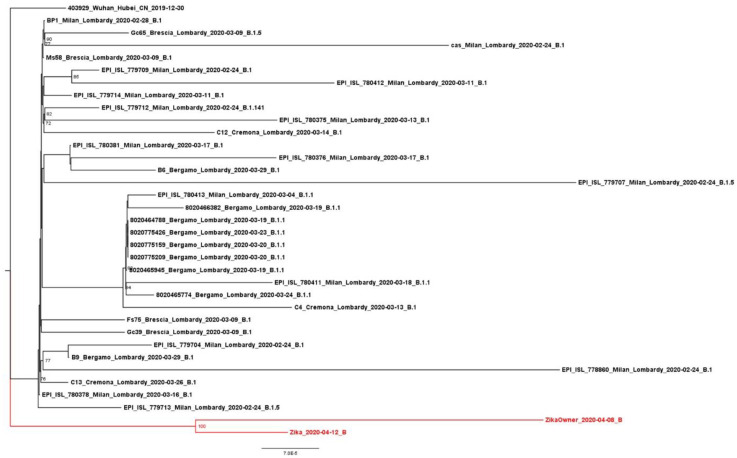
Phylogenetic analysis of whole Italian SARS-CoV-2 genomes showing the relationship between the two strains (in red). The tree was constructed using the maximum likelihood method. The lineage information shown was inferred using Pangolin nomenclature. The reference genome Wuhan-Hu-1 (EPI_ISL_403929) was used as the root of the tree.

**Table 1 pathogens-10-00252-t001:** Lineage classification of the human and feline strains and the identified mutations (Pangolin lineage and Nextclade assignments).

Strain Name	Lineage	Clade	Gene			
			ORF1a	ORF1b	S	N
**Zika_2020-04-12**	B	19A	C357W,S1952L,D4344G	N1830S	D614G,P1213Q	S235Y
**ZikaOwner_2020-04-08**	B	19A	C357W,S1952L,K2446R,D4344G	T17N,T197A,S967P,I1766T,N1830S	V289I,D614G,P1213Q	V244D

## Data Availability

The data presented in this study are available on request from the corresponding author. Full genome sequence will be publicly available on GISAID (https://www.gisaid.org (accessed on 29 May 2020)).

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
