# Peer review of "Human-to-Cat SARS-CoV-2 Transmission: Case Report and Full-Genome Sequencing from an Infected Pet and Its Owner in Northern Italy"

_pathogens, 2021, doi:10.3390/pathogens10020252_

Round 1
Reviewer 1 Report
The present study reports a case of transmission of SARS CoV2-2 from a human to his/her pet cat. The study is interesting and well described, although the novelty of the data is limited because at least 2 previous studies have confirmed the transmission of this virus from human to cats, including sequence studies as presented here. Similar studies with dogs and their owners have also been published in 2020.
Comments:
- Lines 31-34: This is a very long and complicated sentence, I recommend to replace it by “… finding a SARS-CoV-2 seroprevalence of 5.8% in cats, despite their negative results in real time RT-PCR in oropharyngeal…..”
- Line 69: Do you consider that the virus was isolated with Ct 28 after a week of incubation? Did you make a second passage? If not, why? Was the isolate further characterized?
- Line 103: “serologies” should be replace by “serological tests”. Please describe in detail which serological tests were performed with the cat serum. Why did you perform PCR on OPS and RV after 6 months? Were you expecting a positive result after so much time after infection??
- Line 13: it is not so “rarely reported”. Two previous studies have performed genome comparison between viruses in cats and their owners and also in dogs and their owners (doi: 10.1038/s41586-020-2334-5).
- Line 121-122. “a different lineage carried by these strains”?? Difficult to understand. Please describe in more detail your hypothesis to explain why the samples correspond to a completely separate clade with regard to other sequences in Lombardy. This deserves an explanation.
- Line 125: Ref 1 (Decaro et al.) do not say that SARS CoV-2 infection in cats can lead to a severe disease. In fact, there are no studies reporting severe disease in cats due to natural SARS-CoV-2 infection. All the infected cats with no previous preexisting health problems have fully recovered. Please, modify this sentence.
- Line 132: “Privately owned” sounds weird to me. I’d delete it.
- Line 134: also transmission from minks to cats has been confirmed in Netherlands and Denmark. Please include this information and at least one reference.
Author Response
The present study reports a case of transmission of SARS CoV2-2 from a human to his/her pet cat. The study is interesting and well described, although the novelty of the data is limited because at least 2 previous studies have confirmed the transmission of this virus from human to cats, including sequence studies as presented here. Similar studies with dogs and their owners have also been published in 2020.
Comments:
- Lines 31-34: This is a very long and complicated sentence, I recommend to replace it by “… finding a SARS-CoV-2 seroprevalence of 5.8% in cats, despite their negative results in real time RT-PCR in oropharyngeal…..”
Thank you for your comment, the manuscript has been edited for English Language, including rephrasing this particularly hard to read sentence.
- Line 69: Do you consider that the virus was isolated with Ct 28 after a week of incubation? Did you make a second passage? If not, why? Was the isolate further characterized?
A second passage was not performed as time and resources were extremely limited during the first weeks of dramatic escalation of the pandemic in Italy. No further characterizations were made. The remaining material has been stored for further analyses.
- Line 103: “serologies” should be replace by “serological tests”. Please describe in detail which serological tests were performed with the cat serum. Why did you perform PCR on OPS and RV after 6 months? Were you expecting a positive result after so much time after infection??
Serological tests were performed using antibody neutralization technique. A reference (16) was included (lines 104-106).
Regarding the PCR on RV and OPS after six months, as very little is known about the clinical course of SARS-CoV-2 infection in cats, we decided to perform swabs at the same time of venous blood drawing to exclude a possible chronic viral carriage. Unfortunately we couldn’t perform other PCRs before that date because of the dramatic escalation of the pandemic in Italy during spring 2020.
- Line 13: it is not so “rarely reported”. Two previous studies have performed genome comparison between viruses in cats and their owners and also in dogs and their owners (doi: 10.1038/s41586-020-2334-5).
We thank the reviewer for his consideration. We realize that this is not the first case of human-to-pet transmission including WGS. However, in a pandemic situation in which millions of people (and, potentially, their pets) are exposed and/or infected, we think that viral isolation and WGS in both pet and owner is still a relatively rare occurrence.
The text has been however modified using a less strong expression (lines 115-116).
- Line 121-122. “a different lineage carried by these strains”?? Difficult to understand. Please describe in more detail your hypothesis to explain why the samples correspond to a completely separate clade with regard to other sequences in Lombardy. This deserves an explanation.
This part was integrated by Prof.Zehender, reference to a previous work (Lai et al, Viruses, 2020) was added for adjunctive informations on clades circulating in Italy at the time of our work (lines 121-125).
- Line 125: Ref 1 (Decaro et al.) do not say that SARS CoV-2 infection in cats can lead to a severe disease. In fact, there are no studies reporting severe disease in cats due to natural SARS-CoV-2 infection. All the infected cats with no previous preexisting health problems have fully recovered. Please, modify this sentence.
Your consideration is absolutely right, the phrase have been wrongly corrected. The right sentence is “it cannot be excluded, however, that SARS-CoV-2 infection in cats could lead to a severe disease, as it already happens with other feline coronaviruses”. This has been corrected in text (lines 127-128).
- Line 132: “Privately owned” sounds weird to me. I’d delete it.
Thank you, it has been corrected.
- Line 134: also transmission from minks to cats has been confirmed in Netherlands and Denmark. Please include this information and at least one reference.
Thank you for your observation, this information was already present in a previous version, but has been removed due to space requirements. We included it again (lines 31-32).
Reviewer 2 Report
Dear Editor, Authors wrote a very interesting paper and I find it true well wrote and response an important question: transmission human to domestic animal. I believe that One health approach play a key role to COVID burden control. Authors wrote an important paper, high quality and it is not easy to give some suggestions. Below only some minor comments:
- Introduction: no comment
- Case report and virological Analyses: it is very interesting, figure and table help to understand the question. Good job!
- Discussion: Propose some public health actions resulting from your interesting study: ex.A register of infected animals or Prevalence study in animals of infected people to better investigate one health approach and give new possible interpretations and knowledge to the COVID pandemic
Author Response
Dear Editor, Authors wrote a very interesting paper and I find it true well wrote and response an important question: transmission human to domestic animal. I believe that One health approach play a key role to COVID burden control. Authors wrote an important paper, high quality and it is not easy to give some suggestions. Below only some minor comments:
- Introduction: no comment
- Case report and virological Analyses: it is very interesting, figure and table help to understand the question. Good job!
- Discussion: Propose some public health actions resulting from your interesting study: ex.A register of infected animals or Prevalence study in animals of infected people to better investigate one health approach and give new possible interpretations and knowledge to the COVID pandemic
The authors thank the reviewer for his kind words. As of today, at least in Italy, infected or supposedly infected animals are reported trough the health surveillance system, similarly to human subjects. A reference containing suggestions for both pet owners and veterinary services has been added to the paper, including instruction for reporting cases to the World Animal Health Information System (17, line 144).